# Zero-shot Sim2Real Transfer for Magnet-Based Tactile Sensor on Insertion Tasks

**Abstract:** We aim to bridge the sim-to-real gap of dense, distributed, 3-axes, magnet-based tactile sensors, like u-skin, which balances well between tactile density, high-durability, and compactness. Unlike prior works that use binarization techniques, we propose GCS, a novel sim-to-real technique to learn contact-rich insertion skills. We evaluated our approach in *blind* insertion tasks and show zero-shot sim-to-real transfer of RL policies with raw tactile readings as input.

**Keywords:** Tactile Sensor, Sim-to-Real, Insertion

## 1 Introduction

Sense of touch is an important sensing modality for humans. With dense, distributed sensors on our skin, humans can detect subtle changes in external contact, and conduct dexterous skills like tool use. There has been tremendous effort [1, 2, 3, 4, 5, 6, 7, 8, 9, 10, 11, 12, 13, 14, 15, 16, 17] to replicate the capability on robots for manipulation. Among various tactile sensors, magnet-based sensors, like uskin [12, 13], can provide dense, distributed, 3-axes forces, and are durable and compact to be integrated into dexterous hands [18]. However, using magnet-based tactile sensors in robot learning has not been widely explored in previous work.

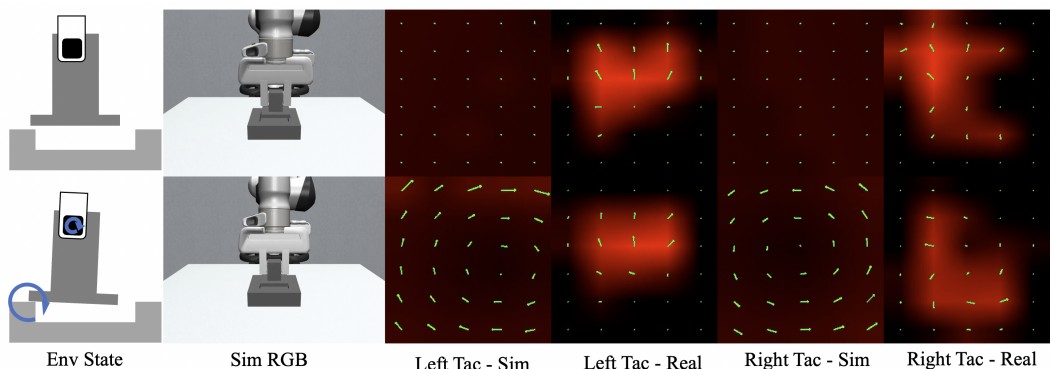

Env State     Sim RGB     Left Tac - Sim    Left Tac - Real    Right Tac - Sim    Right Tac - Real

Figure 1: Comparison of simulated $6 \times 5$ tactile readings in MuJoCo and in the real world under the same physical state. For the right four figures, the redness represents normal forces and the green arrows represent the shear forces, on each taxel. This visualizes the sim-to-real gap of tactile readings.

Our work focuses on bridging the sim-to-real gap of dense, distributed, 3-axes, magnet-based sensors [12, 19]. We use a commercial magnet-based sensor ([19]) with $6 \times 5$ grid of $4.7mm \times 4.7mm$ taxels, i.e., $6 \times 5 \times 3$ tensor readings, for *blind* insertion tasks. Most importantly, we managed to solve the task with a tactile-based RL-policy trained *only* in simulation. Furthermore, in our method, we avoid using techniques like binarization [10, 9], that loses much information from tactile sensors. Instead, we aim to bridge the sim-to-real gap of *raw sensor readings* (Figure 1).

Submitted to the 8th Conference on Robot Learning (CoRL 2024). Do not distribute.

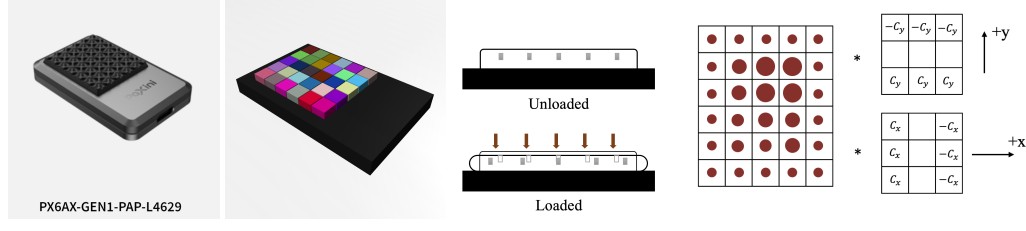

(a) PaXini Sensor     (b) Sensor Model     (c) Poisson Effect     (d) Poisson Convolution

Figure 2: (a) Product PX6AX-GEN1-PAP-L4629 from PaXini Tech, figure from [19]. (b) MJCF visualization of the tactile sensor model in simulation. We use $6 \times 5$ small cubes instead of a single cube. (c) Illustration of Poisson Effect on the tactile sensor in real world. Gray cubes represent the small magnets injected in the gel [12]. (d) Illustration of convolution technique to approximate Poisson Effect noise in real sensor readings. Red dots represent the normal force on each taxel, which is convolved with poisson-effect kernels in both directions. $c_{x,y}$ are hyper-parameters that control the noise scale.

To achieve this, we first identify key aspects of the sim-to-real gap: non-uniform contact, contact Poisson Effect, and force scaling difference. We then develop simple yet efficient techniques to mitigate these gaps (Figure 3): surface with **G**aussian bumps, **C**onvolution for Poisson Effect and domain randomization for Force **S**caling. In our experiment, we show that our approach **GCS** enables *zero-shot* sim-to-real transfer of RL policy on *blind* peg-in-hole insertion tasks, i.e., the robot does not have accurate peg poses. Results suggest that our GCS method outperforms all previous sim-to-real techniques and improves the success rate by **50%**. For some tasks, our RL policy succeeds on 9 out of 10 real-world trials.

## 2   Method

In our work, we use the dense, distributed, 3-axes, magnet-based tactile sensor: PX6AX-GEN1-PAP-L4629 (Figure 2a) from PaXini Tech [19]. To simulate tactile sensors, we extend Robosuite [20] which is built on the MuJoCo [21] physics engine. Similar to Sferrazza et al. [22], we use the Touch Grid plugin provided by MuJoCo to get the aggregate contact forces in each taxel area. In addition, as shown in Figure 2b, instead of using a single cube as the contact pad of the sensor, we discretize the contact pad with small cubes of $4.6mm \times 4.6mm \times 1mm$ (W×H×D).

Figure 1 compares tactile readings in MuJoCo with readings from real world sensors under the same physical state. In the first row, the robot holds the cubic peg handle, which has no contact with external environment. In the second row, the robot holds the same handle, while the peg is in contact with the back rim of the square hole. There exists a significant sim-to-real gap in tactile sensor readings. Please refer to the appendix for an intuitive interpretation of the gap.

### 2.1   GCS: Bridge Sim-to-Real Gap

Prior work bridges the sim-to-real gap via binarization techniques [9, 23, 10]. However, this may lose information that is crucial in other tasks. In our work, we aim to bridge the sim-to-real gap of *raw tactile readings* of dense, distributed, 3-axes tactile forces, as it is the most general form of tactile input. To this end, we propose three simple yet effective techniques to address the above gaps.

**1. Surface with Gaussian Bumps.** We randomize the depth of the small cubes on the surface pad in simulation. Specifically speaking, we randomly select a 2-dimensional Gaussian distribution, i.e., mean $(g_x, g_y) \in \mathcal{G} = [0, 5] \times [0, 6]$ and standard deviation $(s_x, s_y)$. Then, given the coordinate of each taxel $(i, j) \in \mathcal{G}$, we denote $D((i,j), (g_x, g_y)) = \sqrt{(g_x - i)^2/s_x^2 + (g_y - j)^2/s_y^2}$. Then, the depth $h$ of taxel cube $(i, j)$ is

$$h(i, j) = h_{\min} + \frac{D((i,j), (g_x, g_y))}{\max_{(i,j) \in \mathcal{G}} D((i,j), (g_x, g_y))} * (h_{\max} - h_{\min}).$$

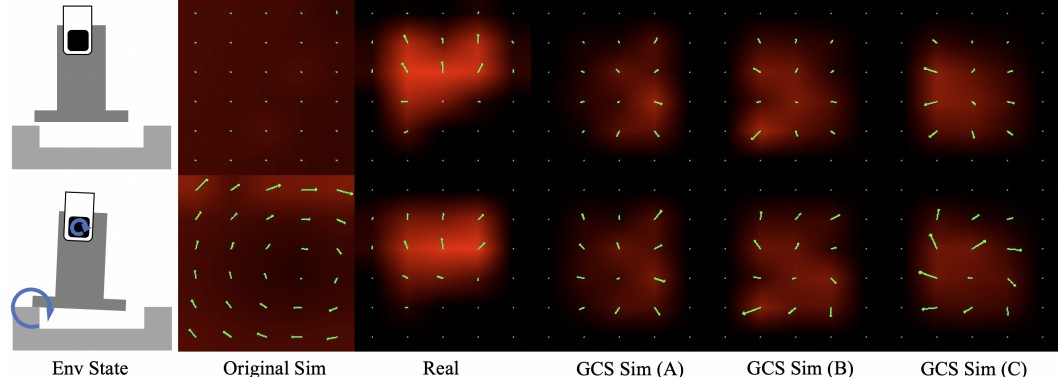

Figure 3: For right 3 figures, we show the *left* finger GCS tactile readings (A)(B)(C) under 3 random parameters sampled from Table 3, and compare them with the real-world tactile readings and the original raw simulation tactile readings under the same physical states as in Figure 1.

Here $h_{\max}$ and $h_{\min}$ denotes the maximal and minimal cube depth, which are chosen as hyperparameters.

**2. Convolution for Poisson Effect.** We approximate the Poisson Effect noise via a simple approach, avoiding time-consuming methods like FEM simulation. Observing that the scale of the shear force is proportional to the normal force in its surroundings, we approximate the Poisson Effect shear displacement by convolving the normal forces tensor with poisson-effect kernels, illustrated in Figure 2d. The poisson-effect scale is determined by parameters $c_x, c_y$ respectively for each direction.

**3. Domain Randomization of Force Scaling.** We randomize the force scaling factors in each episode during training. To be specific, we randomize the scaling for each axis of the all taxels, i.e., $\alpha_k, k \in [x, y, z]$, and independently randomize the scale of each taxel $\beta_{ij}, (i, j) \in \mathcal{G}$. Namely, each taxel reading $f_k(i, j) = \hat{f}_k(i, j) * \alpha_k * \beta_{ij}$, where $\hat{f}_k(i, j)$ is the raw force reading from the physics engine for taxel $(i, j)$ in direction $k$.

Figure 3 illustrates the revised tactile readings with GCS, in the same physical states as Figure 1.

# 3 Experiment

**Task.** We use the following 6 *blind* insertion tasks in our experiment (Figure 6). The robot is *blind* so that it does not know the pose of the peg, while the peg is randomly placed between fingers at each initialization. Consequently, the robot is required to infer contact between the peg and the hole base from tactile sensing. Please refer to the appendix for task details.

- **RY-2mm**: Round peg with cylinder handle, and 2mm clearance round hole.
- **RU-2mm**: Round peg with cubic handle, and 2mm clearance round hole.
- **SQ-2mm**: Square peg with cubic handle, and 2mm clearance square hole.
- **SQ-1mm**: Square peg with cubic handle, and 1mm clearance square hole.
- **SX-2mm**: Square peg with cubic handle, and 2mm clearance groove along the x-axis. In addition, the hole pose is also blind to the robot.
- **SY-2mm**: Square peg with cubic handle, and 2mm clearance groove along the y-axis. In addition, the hole pose is also blind to the robot.

**Baselines.** We compare GCS with the following baselines, that are trained with asymmetric SAC. Implementation details are listed in the appendix.

1. **No Tactile (NT).** The policy $\pi(p_t)$ is a 2-layer MLP that does not use tactile information.
2. **Total Force (TF).** Total contact force $f_t$ on each surface pad is computed by summing all taxel readings. The force input $f_t$ is a $6k$-dim vector, which is concatenated with $p_t$, i.e., $\pi(p_t, f_t)$.
3. **Taxel Binarization (TB).** Similar as [23, 9], we binarize the taxel reading with a 0.1N threshold. The tactile input is a $2k \times 6 \times 5$ tensor of $[0, 1]$. We encode with a similar CNN.

4. **Taxel Direction (TD).** Inspired by [10], we binarize each taxel reading in each axis with 0.1N threshold, together with the sign to represent the direction. Thus, the tactile input is a $6k \times 6 \times 5$ tensor of value $[-1, 0, 1]$, encoded with a similar CNN.

5. **Direct Transfer (DT).** We use the tactile reading from the simulator (Figure 1) directly, which is a $6k \times 6 \times 5$ tensor, and is encoded with a similar CNN encoder.

**Results.** For real-world evaluation, we test each method with 10 randomly initialized trials. In simulation, we test with 50 trials. The result is shown in Table 1.

Table 1: Comparative evaluation on success rate of baselines and ours GCS in simulation and real world. NT: No Tactile; TF: Total Force; TB: Taxel Binarization; TD: Taxel Direction; DT: Direct Transfer.

| Method | RY-2mm | RU-2mm | SQ-2mm | SQ-1mm | SX-2mm | SY-2mm | Avg |
|---|---|---|---|---|---|---|---|
| NT (Sim) | 0.94 | 0.82 | 0.6 | 0.56 | 0.42 | 0.2 | 0.59 |
| NT (Real) | 0.4 | 0.4 | 0.0 | 0.1 | 0.0 | 0.0 | 0.15 |
| TF (Sim) | 1.0 | 0.98 | 0.8 | 0.0 | 1.0 | 0.74 | 0.75 |
| TF (Real) | 0.4 | 0.3 | 0.3 | 0.0 | 0.4 | 0.1 | 0.25 |
| TB (Sim) | 1.0 | 0.82 | 0.62 | 0.36 | 0.84 | 0.1 | 0.62 |
| TB (Real) | 0.5 | 0.3 | 0.4 | 0.1 | 0.5 | 0.2 | 0.33 |
| TD (Sim) | 1.0 | 0.86 | 0.9 | 0.88 | 0.88 | 0.80 | 0.88 |
| TD (Real) | 0.1 | 0.4 | 0.0 | 0.2 | 0.2 | 0.2 | 0.18 |
| DT (Sim) | 1.0 | 0.96 | 0.84 | 0.94 | 0.96 | 0.84 | 0.92 |
| DT (Real) | 0.6 | 0.2 | 0.1 | 0.3 | 0.4 | 0.1 | 0.28 |
| GCS (Sim) | 1.0 | 0.9 | 0.84 | 0.98 | 0.98 | 0.76 | 0.91 |
| GCS (Real) | **0.9** | **0.8** | **0.9** | **0.6** | **0.9** | **0.7** | **0.8** |

We find that baselines cannot produce good policies that transfer well to the real world. For NT, it has only <60% success rate on square peg tasks even in simulation and in real world, it has <10% success rate. For TF, it has a higher success rate in simulation than NT. However, as discussed in Section 2, the real-world reading is quite noisy. We observe that the total force on the pad can even point upward when there is no external contact. Consequently, the policy cannot perform well in real world.

Binarization techniques (TB, TD) are useful for in-hand orientation tasks [9, 23] translation [10], we find that they are not useful for insertion tasks. The real-world shear force noise makes the feature in these methods non-transferrable to the real-world rollouts. In terms of DT, owing to large sim-to-real gap (Figure 1row), the real-world success rate also is low due to distribution shift.

We observe that GCS outperforms all baselines by a 50% margin on average. In many tasks such as RY-2mm, SQ-2mm, and SX-2mm, it has a real-world success rate of 90% with barely any performance drop than in the simulation. The largest sim-to-real gap occurs in the SQ-1mm case, which is the hardest task owing to the small clearance. On SQ-1mm, one needs fine-grained and compliant control. Small action error, potentially from sim-to-real tactile gap, will cause the peg to miss the hole in the next step.

# 4 Conclusion

Dense, distributed, 3-axes, magnet-based tactile sensor (i.e., u-skin) balances well between durability, tactile density, and compactness. In our work, we try to bridge the sim-to-real gap of these sensors. Here, we identify three aspects of the sim-to-real gap: non-uniform contact, poisson effect, and scale difference, and propose a novel GCS method to mitigate these gaps. We evaluate our approach on *blind* insertion tasks and show that GCS improves the real-world success rate by 50% over baselines. Our work creates new possibilities to future sim-to-real learning of visual-tactile manipulation skills with magnet-based sensors.

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

## A Related Works

**Robot Manipulation with Tactile Sensors.** Tactile sensors are crucial for many manipulation tasks. They provide information of contact between robots and objects, and contact between objects in hand and its surroundings. Moreover, such information cannot be easily revealed from visual information alone. Many prior works used vision-based optical tactile sensors (e.g., Gelsight [1, 2]) in grasping [3, 4], slip detection [24], cable manipulation [5, 6], cloth manipulation [25], visual-tactile slam [7, 8], contact-rich insertion [26, 27, 28], in hand manipulation [9] and in exploiting external contact [29, 30]. Recently, works have also used piezoelectric and magnet-based sensors [14, 31, 32] for various manipulation tasks, including visual-tactile in-hand reorientation and translation [23, 33, 10]. Most importantly, a piezoelectric sensor, i.e., 3D-ViTac [11, 34] and a magnet-based sensor, e.g., u-skin [12, 13] have been introduced to robot manipulation. These two sensors can provide dense, distributed tactile signals. Works have used these high-dimensional tactile information for complex manipulation tasks via imitation learning on tele-operated demonstrations [11, 35, 36, 37, 38]. Unlike these prior work, we focus on bridging the sim-to-real gap of dense, distributed, 3-axes, magnet-based sensors.

**Tactile Sensor Sim-to-Real.** There is only limited data involving real-world tactile readings [39], e.s.p., data collected during robot manipulation [29, 11]. In contrast, simulation data are relatively cheap to acquire and work has shown learning dexterous skills with only simulation data [40, 9, 23]. However, as the mechanisms of the rigid-body physics engines [21, 41] are different from the real world physics, bridging the large sim-to-real gap for manipulation with tactile sensors remains a challenge. For optical tactile sensors, prior work developed example-based simulators [42, 43], differentiable simulators [44], rigid-body simulators coupled with penalty-based models [17, 45] and FEM-based simulators [46, 47, 48]. Notably, Chen et al. [47] achieved zero-shot sim-to-real transfer on contact-rich tasks [47] with IPC-based [49] simulation. In addition, works have also shown success on dexterous in-hand orientation [23, 9] and translation [10], with binary tactile signals from magnet-based or piezoelectric sensors. However, binarization loses a lot of information from dense, distributed, 3-axes sensors. For example, in peg-in-hole insertion tasks, binarization does not provide information of contact between the peg and the hole base. In our work, we use the raw tactile reading directly as input to the RL policy and we develop a different technique to bridge the sim-to-real gap of raw tactile reading for dense, distributed, 3-axes, magnet-based sensors.

## B Additional Results

### B.1 Ablation Study

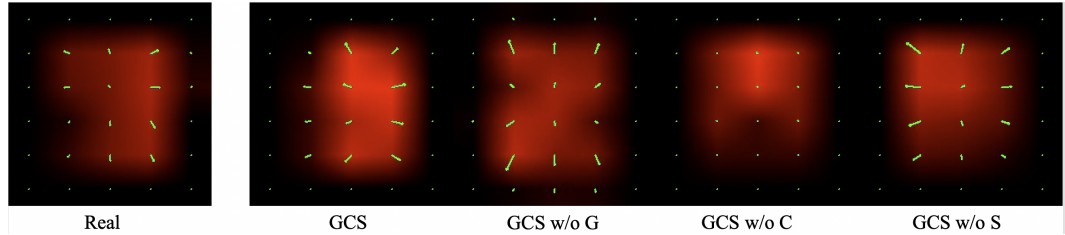

Figure 4: Visualization of ablation's left tactile readings in the no contact state. Noticing that readings in GCS and ablations are randomized in training.

Here, we demonstrate the necessity of all *G, C, S* technique to bridge the sim-to-real gap.

Table 2 shows the real-world success rate of the ablation. In simulation, the success rate is >80% for all ablations. Clearly, results show that all three *G,C,S* techniques are necessary. This is especially the case in SX-2mm and SY-2mm tasks, where it only uses tactile reading as input. Consequently, distribution mismatch between simulated and real-world tactile readings will result in the failure of the contact-rich insertion task. Figure 4 visualizes the tactile reading in simulation of different ablations. With all *G,C,S* techniques, the reading looks more similar to real world.

Table 2: Ablation study of GCS on SX-2mm, SY-2mm, and SQ-2mm tasks. In each ablation, we remove one technique in Section 2.1.

|        | GCS | GCS w/o G | GCS w/o S | GCS w/o C |
|--------|-----|-----------|-----------|-----------|
| SX-2mm | 0.9 | 0.4       | 0.3       | 0.3       |
| SY-2mm | 0.7 | 0.0       | 0.6       | 0.2       |
| SQ-2mm | 0.9 | 0.9       | 0.8       | 0.6       |

## B.2 Policy Interpretation

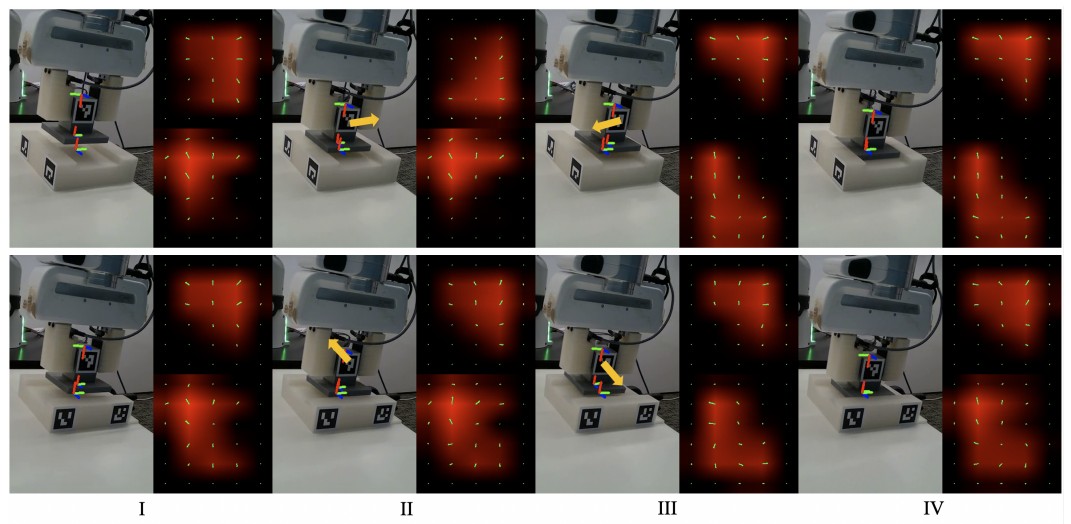

Figure 5: Examples of policy rollout in SX-2mm (first row) and SY-2mm (second row). Stage I corresponds to the initial state, and Stage IV corresponds to the success state. Stage II and Stage III corresponds to the critical contact states that the policy infers the relative position from (Section B.2).

To better understand how the policy trained in simulation transfers to real world, we visualize trajectories in SX-2mm and SY-2mm tasks, in Figure 5. In SY-2mm, it detects the relative position between the peg and the hole by the change of the shear forces. In stage II, the peg contacted with the front rim and thus shear force markers on the left finger changed counterclockwise and clockwise on the right finger sensor. Consequently, the policy commanded the hand to move backward. However, it overshoot and contacted the back rim (stage III). Again, it detected the opposite change on both sensors and thus moved forward. This time, it managed to insert the peg in the groove.

For SX-2mm, it detects the relative position by the change of distribution of normal force on the sensor pads. In stage II, it contacted the left rim and the normal force shifted downward on the left sensor, owing to the external torque. The policy then commanded it to move right. In stage III, it contacted with the right rim and the normal force on the right pad shifted downward this time. Then it moved left in fine-grained steps until the peg was inserted into the groove.

## C  Experiment Details

**Sim-to-Real Gap.** We identify that the gap comes from three main aspects:

1. **Non-uniform Contact**. The actual surface gel is uneven and the injected small magnets [12] are not completely symmetric in terms of magnetic field. Therefore, we observe non-uniform activation of the dense distributed Hall Effect sensors, i.e., non-uniform normal force readings, despite that the contact surface is flat.

2. **Contact Poisson Effect**. There exists a significant non-zero shear force, even when the peg is not in contact with the environment. We find that this results from the Poisson Effect of the contact gel, which is illustrated in Figure 2c.

3. **Different Force Scale**. We find that the absolute scale of real-world sensor force differs from the scale in simulation. This comes from the difference between the physics engine and real-world physics.

**Randomization Ranges** Table 3 summarizes the randomization range of all factors in our pipeline.

Table 3: Domain Randomization (DR) distribution of key parameters in our GCS method.

| Parameters | Axes Force $\alpha_k$ | Taxel Force $\beta_{ij}$ | Convolution $c_{x,y}$ | Deviation $s_{x,y}$ |
|---|---|---|---|---|
| DR Range | $\mathcal{U}(0.5, 1.5)$ | $\mathcal{N}(1.0, 0.25), 20\%$ Dropout | $\mathcal{U}(0.1, 0.3)$ | $\mathcal{U}(1.0, 3.0)$ |

**Task.** We use Operation Space Controller (OSC) at 20Hz. For RY-2mm, RU-2mm, SQ-2mm, SQ-1mm, the policy network outputs 3-dim position delta (XYZ) in $[-1, 1]$, scaled with 1cm, except for SQ-1mm, which is scaled with 5mm. Here, we freeze the rotation in OSC to simplify the tasks. For SX-2mm, we further freeze X-axis and for SY-2mm, we freeze Y-axis.

At initialization of each episode in simulation, we randomize the parameters in Table 3. Similar to [50], we move the gripper to the peg grasp pose with small randomization before the policy rollout. In real world, we first command a randomized initial fingertip pose. Then we place the peg in the gripper with small position randomization.

For real-world evaluation, we will consider success insertion if the bottom of the peg is in the hole. Each trial is truncated at 120 steps for SX-2mm and SY-2mm, and 200 steps for RY-2mm, RU-2mm, SQ-2mm, and SQ-1mm.

**Reward.** The RL reward for all tasks is defined with the following 4 terms.

- *Reaching Reward.* We compute the average distance $d$ of keypoints of the peg to correspondent points on base. The reaching reward is $r_{\text{reach}} = \frac{1}{10*d+\epsilon}$.
- *Engagement Reward.* Given the engagement distance $h$ (the distance between the bottom of peg to the base bottom), if the peg is engaged with the hole, then we have $r_{\text{engage}} = \frac{1}{10*h+\epsilon}$, otherwise $r_{\text{engage}} = 0$.
- *Success Bonus.* The success bonus $r_{\text{succ}}$ if $h < h_{\text{th}}$, where $h_{\text{th}}$ is a threshold value.
- *Action Penalty.* We penalize the action scale with $r_{\text{action}} = - \parallel a \parallel^2$.

The total reward is the weighted sum of these 4 terms. The weights are selected as hyper-paramters.

**Hardware.** We custom 3d-printed fingers to attach sensors to the Franka Research 3 Gripper (Figure 6). We place Realsense L515 in front of the scene to capture object poses, which are labelled with Aruco markers. All poses are computed relative to the robot base frame with camera calibration. Noticing that peg poses are not available to the robot and base poses in SX-2mm and SY-2mm are also not available. Here, we put markers for tracking and task initialization. We use Deoxys [51] to run the RL policy with Operation Space Controller (OSC) on the real robot.

**RL Policy.** For all policies, the input includes relative pose from robot fingertip to the hole base $p_t$ (zeros for SX-2mm, SY-2mm tasks). For GCS and baselines (Section **??**) with tactile inputs, we use a history of $k$ tactile readings, where $k$ is a hyperparameter for each method. In GCS, the tactile input $o_t$ is a $6k \times 6 \times 5$ ($C \times H \times W$) tensor. The policy network first encodes the tactile reading with a CNN to a 128-dim latent vector $t_t = T(o_t)$ and encodes $h_t = E(p_t)$ to a 128-dim vector. Then, we feed them into a 2-layer MLP action head $a_t = \pi(E(p_t), T(o_t))$. Here, we use asymmetric SAC [52] to train the RL policy in Robosuite [20], implemented based on StableBaseline3 [53]. The critic network takes ground truth pose of objects and fingertip poses as input, but without tactile readings. For all tasks, the reward is the sum of reaching reward, engagement reward, and success bonus, which is similar to [50]. Please refer to the appendix for RL training details.

**Training.** We select the optimal $k$ in $\{1, 5, 10, 20\}$ for all baselines. We choose k=1 for TF, k=10 for TB, TD and DT. For GCS, we choose k=20 for all tasks except SY-2mm, which we find k=10 works better. In Section B.1, GCS ablations use the same $k$ as GCS.

The tactile tensor encoder $T$ in TB, TD, DT, and GCS is a 4-layer CNN with BatchNorm and ReLU activation and the encoder for $p_t$ is a single-layer MLP. All MLP policy heads are 256-dim, 2-layer MLPs. For the critic in SAC, it is a 2-layer MLP that takes relative pose from peg to base and end-effector to base as inputs. The implementation is based on StableBaseline3 [53].

## D   Additional Figures

**Task Illustration.** Figure 6 shows the illustration examples of our experiments.

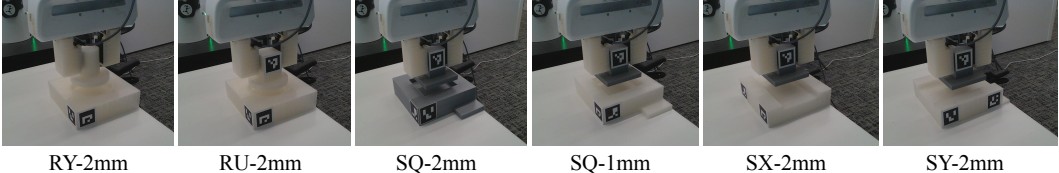

| RY-2mm | RU-2mm | SQ-2mm | SQ-1mm | SX-2mm | SY-2mm |

Figure 6: Illustration of all 6 *blind* insertion tasks in our experiment.

**Sensor Finger Mount.** Figure 7a and Figure 7b show the 3d-printed custom finger to mount the tactile sensor and its CAD model. We attach the sensor with double-sided tape on the fingers.

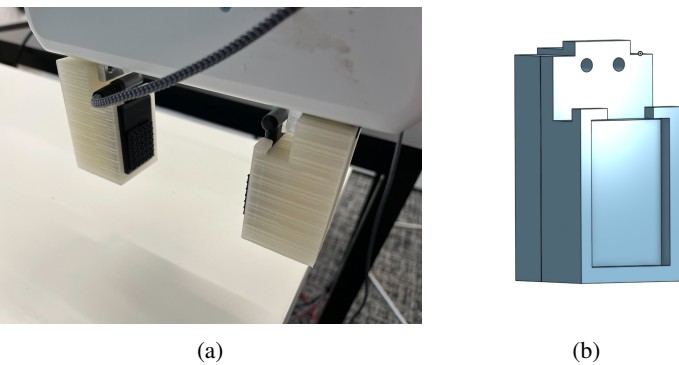

(a)                                      (b)

Figure 7: (a) The finger mount to attach the sensor on the Franka gripper. (b) CAD model for the custom finger.

**IsaacGym Tactile Reading.** Figure 8 shows the tactile rendering in Isaac-Gym [41] holding the square peg, with no contact with the external environment. The sensor model is similar as Figure 2b. Clearly, the tactile reading is noisy and unstructured. Therefore, we choose MuJoCo as our base simulator.

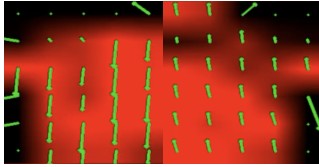

Figure 8: Tactile rendering in Isaac-Gym [41] with square peg at no contact state.

**Tactile Reading in RY-2mm.** RY-2mm use a round peg with cylindric handle. Consequently, the contact surface is quite narrow between. Thus, RY-2mm is a relatively easy task as certain pose information can be revealed from the tactile reading. Figure 9 compares GCS readings with real-world readings.

**Policy Interpretation on SQ-1mm.** With 1mm clearance, SQ-1mm is quite difficult and it requires soft, compliant motion. Figure 10 visualizes a policy rollout in SQ-1mm. In Stage II, the peg contacted with the back rim, which the policy managed to infer from small shear force change. Thus, it lifted the peg and moved forward. However, it overshoot and resulted in contact with the front rim (Stage III). This time, the policy lifted the peg and halt for a while instead of moving backward directly (Stage IV). Then it started to push the peg down, but it collided with the right rim and resulted in change in the distribution of normal force (Stage V). Then it moved left and downward. Here, it used very small actions and the peg was inserted successfully.

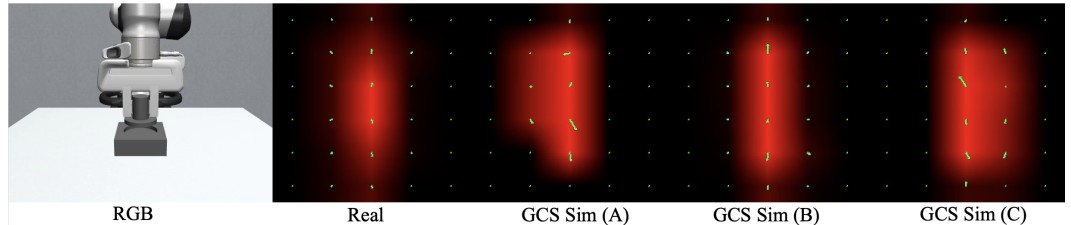

RGB           Real           GCS Sim (A)           GCS Sim (B)           GCS Sim (C)

Figure 9: Tactile reading of RY-2mm task. The peg does not have contact with the base.

**Videos.** We have included example videos of real-world policy rollouts in the appendix.

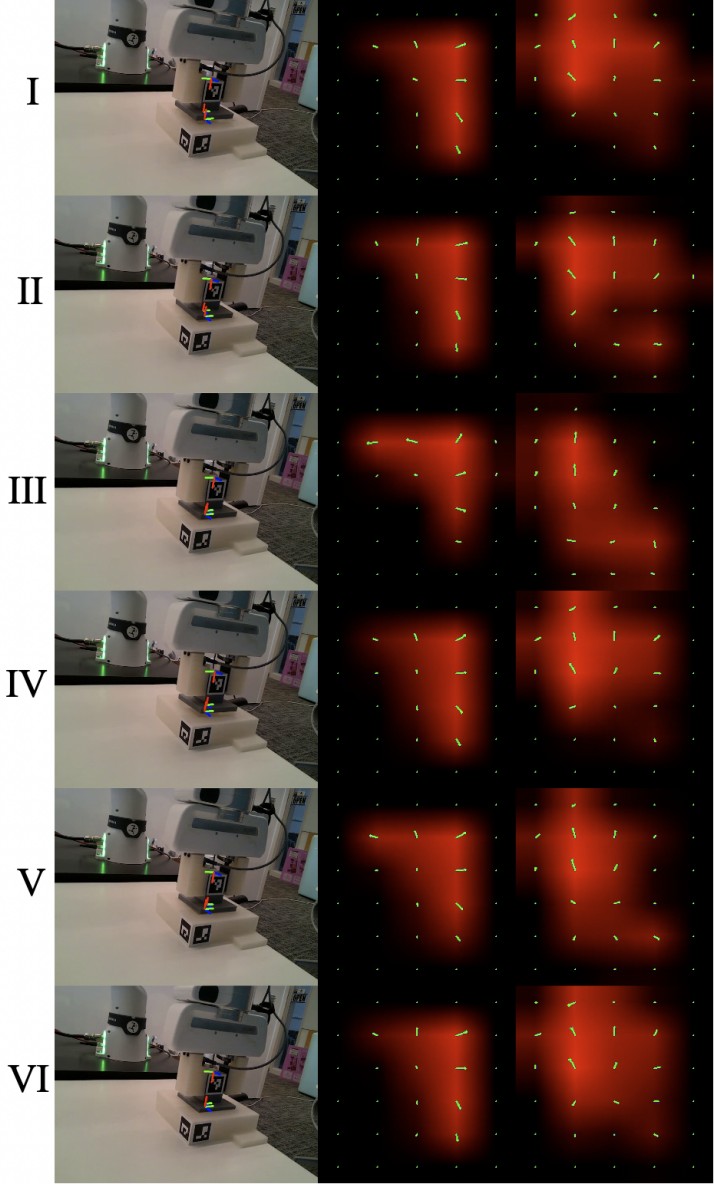

Figure 10: Policy rollout on SQ-1mm task.

