# OpenReview forum: "Zero-shot Sim2Real Transfer for Magnet-Based Tactile Sensor on Insertion Tasks"
_robot-learning.org/CoRL/2025/Workshop/Dexterous_Manipulation — CoRL 2025 Workshop Dexterous Manipulation Spotlight_

### Official Review · Reviewer_FWqq · 2025-09-15

**Rating:** 8
**Confidence:** 4

**Review:**

Summary:
This paper proposes GCS, a sim-to-real technique for 3-axis magnet-based tactile sensors. Compared to previous approaches that binarize the contact signal, GCS preserves rich tactile information while showing strong sim-to-real transfer.
Strengths:
Novelty: Sim-to-real transfer of rich tactile signals is an important feature, and the method the authors proposed is novel. The intuition of the approach is also clear, motivated from the difference between tactile signals in sim and real.
Performance: GCS shows stronger sim-to-real performance compared to other ways of leveraging tactile signals.
Weaknesses
Experiment
It would be great to see more experiments on other type of contact rich tasks, such as in-hand reorientation.
It is unclear how the authors experimented with other baselines. For instance, it is common to randomize the physics of the simulator for sim-to-real, and it is unclear whether the authors included this for the experiment. If not, comparing such domain randomization and the authors’ approach would be desirable.
Questions:
Is ‘no tactile’ approach using state information (i.e. pose of the peg) or vision?
Why is the sim-to-real gap for ‘no tactile’ and ‘binarized tactile signal’ so large?
Did the authors consider any type of domain randomization for physics of the simulator?

---

### Decision · Program_Chairs · 2025-09-18

Accept (Spotlight)